# Accelerated consensus via Min-Sum Splitting

**Patrick Rebeschini**
Department of Statistics
University of Oxford
patrick.rebeschini@stats.ox.ac.uk

**Sekhar Tatikonda**
Department of Electrical Engineering
Yale University
sekhar.tatikonda@yale.edu

## Abstract

We apply the Min-Sum message-passing protocol to solve the consensus problem in distributed optimization. We show that while the ordinary Min-Sum algorithm does not converge, a modified version of it known as Splitting yields convergence to the problem solution. We prove that a proper choice of the tuning parameters allows Min-Sum Splitting to yield subdiffusive accelerated convergence rates, matching the rates obtained by shift-register methods. The acceleration scheme embodied by Min-Sum Splitting for the consensus problem bears similarities with lifted Markov chains techniques and with multi-step first order methods in convex optimization.

## 1 Introduction

Min-Sum is a local message-passing algorithm designed to distributedly optimize an objective function that can be written as a sum of component functions, each of which depends on a subset of the decision variables. Due to its simplicity, Min-Sum has emerged as canonical protocol to address large scale problems in a variety of domains, including signal processing, statistics, and machine learning. For problems supported on tree graphs, the Min-Sum algorithm corresponds to dynamic programming and is guaranteed to converge to the problem solution. For arbitrary graphs, the ordinary Min-Sum algorithm may fail to converge, or it may converge to something different than the problem solution [28]. In the case of strictly convex objective functions, there are known sufficient conditions to guarantee the convergence and correctness of the algorithm. The most general condition requires the Hessian of the objective function to be scaled diagonally dominant [28, 25]. While the Min-Sum scheme can be applied to optimization problems with constraints, by incorporating the constraints into the objective function as hard barriers, the known sufficient conditions do not apply in this case.

In [34], a generalization of the traditional Min-Sum scheme has been proposed, based on a reparametrization of the original objective function. This algorithm is called Splitting, as it can be derived by creating equivalent graph representations for the objective function by "splitting" the nodes of the original graph. In the case of unconstrained problems with quadratic objective functions, where Min-Sum is also known as Gaussian Belief Propagation, the algorithm with splitting has been shown to yield convergence in settings where the ordinary Min-Sum does not converge [35]. To date, a theoretical investigation of the rates of convergence of Min-Sum Splitting has not been established.

In this paper we establish rates of convergence for the Min-Sum Splitting algorithm applied to solve the consensus problem, which can be formulated as an equality-constrained problem in optimization. The basic version of the consensus problem is the network averaging problem. In this setting, each node in a graph is assigned a real number, and the goal is to design a distributed protocol that allows the nodes to iteratively exchange information with their neighbors so to arrive at consensus on the average across the network. Early work include [42, 41]. The design of distributed algorithms to solve the averaging problem has received a lot of attention recently, as consensus represents a widely-used primitive to compute aggregate statistics in a variety of fields. Applications include, for instance, estimation problems in sensor networks, distributed tracking and localization, multi-agents coordination, and distributed inference [20, 21, 9, 19]. Consensus is typically combined with some

form of local optimization over a peer-to-peer network, as in the case of iterative subgradient methods [29, 40, 17, 10, 6, 16, 39]. In large-scale machine learning, consensus is used as a tool to distribute the minimization of a loss function over a large dataset into a network of processors that can exchange and aggregate information, and only have access to a subset of the data [31, 11, 26, 3].

Classical algorithms to solve the network averaging problem involve linear dynamical systems supported on the nodes of the graph. Even when the coefficients that control the dynamics are optimized, these methods are known to suffer from a "diffusive" rate of convergence, which corresponds to the rate of convergence to stationarity exhibited by the "diffusion" random walk naturally associated to a graph [44, 2]. This rate is optimal for graphs with good expansion properties, such as complete graphs or expanders. In this case the convergence time, i.e., the number of iterations required to reach a prescribed level of error accuracy $\varepsilon > 0$ in the $\ell_2$ norm relative to the initial condition, scales independently of the dimension of the problem, as $\Theta(\log 1/\varepsilon)$. For graphs with geometry this rate is suboptimal [7], and it does not yield a convergence time that matches the lower bound $\Omega(D \log 1/\varepsilon)$, where $D$ is the graph diameter [37, 36]. For example, in both cycle graphs and in grid-like topologies the number of iterations scale like $\Theta(D^2 \log 1/\varepsilon)$ (if $n$ is the number of nodes, $D \sim n$ in a cycle and $D \sim \sqrt{n}$ in a two-dimensional torus). $\Theta(D^2 \log 1/\varepsilon)$ is also the convergence time exhibited in random geometric graphs, which represent the relevant topologies for many applications in sensor networks [9]. In [7] it was established that for a class of graphs with geometry (polynomial growth or finite doubling dimension), the mixing time of any reversible Markov chain scales at least like $D^2$, embodying the fact that symmetric walks on these graphs take $D^2$ steps to travel distances of order $D$.

Min-Sum schemes to solve the consensus problem have been previously investigated in [27]. The authors show that the ordinary Min-Sum algorithm does not converge in graphs with cycles. They investigate a modified version of it that uses a soft barrier function to incorporate the equality constrains into the objective function. In the case of $d$-regular graphs, upon a proper choice of initial conditions, the authors show that the algorithm they propose reduces to a linear process supported on the directed edges of the graph, and they characterize the convergence time of the algorithm in terms of the Cesàro mixing time of a Markov chain defined on the set of directed edges of the original graph. In the case of cycle graphs (i.e., $d = 2$), they prove that the mixing time scales like $O(D)$, which yields the convergence time $O(D/\varepsilon \log 1/\varepsilon)$. See Theorem 4 and Theorem 5 in [27]. In the case of $(d/2)$-dimensional tori ($D \sim n^{2/d}$), they conjecture that the mixing time is $\Theta(D^{2(d-1)/d})$, but do not present bounds for the convergence time. See Conjecture 1 in [27]. For other graph topologies, they leave the mixing time (and convergence time) achieved by their method as an open question.

In this paper we show that the Min-Sum scheme based on splitting yields convergence to the consensus solution, and we analytically establish rates of convergence for *any* graph topology. First, we show that a certain parametrization of the Min-Sum protocol for consensus yields a linear message-passing update for *any* graph and for *any* choice of initial conditions. Second, we show that the introduction of the splitting parameters is not only fundamental to guarantee the convergence and correctness of the Min-Sum scheme in the consensus problem, but that proper tuning of these parameters yields accelerated (i.e., "subdiffusive") asymptotic rates of convergence. We establish a square-root improvement for the asymptotic convergence time over diffusive methods, which allows Min-Sum Splitting to scale like $O(D \log(D/\varepsilon))$ for cycles and tori. Our results show that Min-Sum schemes are competitive and get close to the optimal rate $O(D \log(1/\varepsilon))$ recently established for some algorithms based on Nesterov's acceleration [30, 36]. The main tool used for the analysis involves the construction of an auxiliary linear process supported on the nodes of the original graph to track the evolution of the Min-Sum Splitting algorithm, which is instead supported on the directed edges. This construction allows us to relate the convergence time of the Min-Sum scheme to the spectral gap of the matrix describing the dynamics of the auxiliary process, which is easier to analyze than the matrix describing the dynamics on the edges as in [27].

In the literature, overcoming the suboptimal convergence rate of classical algorithms for network averaging consensus has motivated the design of several accelerated methods. Two main lines of research have been developed, and seem to have evolved independently of each others: one involves lifted Markov chains techniques, see [37] for a review, the other involves accelerated first order methods in convex optimization, see [13] for a review. Another contribution of this paper is to show that Min-Sum Splitting bears similarities with both types of accelerated methods. On the one hand, Min-Sum can be seen as a process on a lifted space, which is the space of directed edges in the original graph. Here, splitting is seen to introduce a directionality in the message exchange of the ordinary Min-Sum protocol that is analogous to the directionality introduced in non-reversible

random walks on lifted graphs to achieve faster convergence to stationarity. The advantage of the Min-Sum algorithm over lifted Markov chain methods is that no lifted graph needs to be constructed. On the other hand, the directionality induced on the edges by splitting translates into a memory term for the auxiliary algorithm running on the nodes. This memory term, which allows nodes to remember previous values and incorporate them into the next update, directly relates the Min-Sum Splitting algorithm to accelerated multi-step first order methods in convex optimization. In particular, we show that a proper choice of the splitting parameters recovers the same matrix that support the evolution of shift-register methods used in numerical analysis for linear solvers, and, as a consequence, we recover the same accelerated rate of convergence for consensus [45, 4, 24].

To summarize, the main contributions of this paper are:

1. First connection of Min-Sum schemes with lifted Markov chains techniques and multi-step methods in convex optimization.

2. First proof of how the directionality embedded in Belief Propagation protocols can be tuned and exploited to accelerate the convergence rate towards the problem solution.

3. First analysis of convergence rates for Min-Sum Splitting. New proof technique based on the introduction of an auxiliary process to track the evolution of the algorithm on the nodes.

4. Design of a Min-Sum protocol for the consensus problem that achieves better convergence rates than the ones established (and conjectured) for the Min-Sum method in [27].

Our results motivate further studies to generalize the acceleration due to splittings to other problems.

The paper is organized as follows. In Section 2 we introduce the Min-Sum Splitting algorithm in its general form. In Section 3 we describe the consensus problem and review the classical diffusive algorithms. In Section 4 we review the main accelerated methods that have been proposed in the literature. In Section 5 we specialize the Min-Sum Splitting algorithm to the consensus problem, and show that a proper parametrization yields a linear exchange of messages supported on the directed edges of the graph. In Section 6 we derive the auxiliary message-passing algorithm that allows us to track the evolution of the Min-Sum Splitting algorithm via a linear process with memory supported on the nodes of the graph. In Section 7 we state Theorem 1, which shows that a proper choice of the tuning parameters recovers the rates of shift-registers. Proofs are given in the supplementary material.

## 2 The Min-Sum Splitting algorithm

The Min-Sum algorithm is a distributed routine to optimize a cost function that is the sum of components supported on a given graph structure. Given a simple graph $G = (V, E)$ with $n := |V|$ vertices and $m := |E|$ edges, let us assume that we are given a set of functions $\phi_v : \mathbb{R} \to \mathbb{R} \cup \{\infty\}$, for each $v \in V$, and $\phi_{vw} = \phi_{wv} : \mathbb{R} \times \mathbb{R} \to \mathbb{R} \cup \{\infty\}$, for each $\{v, w\} \in E$, and that we want to solve the following problem over the decision variables $x = (x_v)_{v \in V} \in \mathbb{R}^V$:

$$\text{minimize} \quad \sum_{v \in V} \phi_v(x_v) + \sum_{\{v,w\} \in E} \phi_{vw}(x_v, x_w). \tag{1}$$

The Min-Sum algorithm describes an iterative exchange of *messages*—which are functions of the decision variables—associated to each *directed* edge in $G$. Let $\mathcal{E} := \{(v, w) \in V \times V : \{v, w\} \in E\}$ be the set of directed edges associated to the undirected edges in $E$ (each edge in $E$ corresponds to two edges in $\mathcal{E}$). In this work we consider the synchronous implementation of the Min-Sum algorithm where at any given time step $s$, each directed edge $(v, w) \in \mathcal{E}$ supports two messages, $\hat{\xi}^s_{vw}, \hat{\mu}^s_{vw} : \mathbb{R} \to \mathbb{R} \cup \{\infty\}$. Messages are computed iteratively. Given an initial choice of messages $\hat{\mu}^0 = (\hat{\mu}^0_{vw})_{(v,w) \in \mathcal{E}}$, the Min-Sum scheme that we investigate in this paper is given in Algorithm 1. Henceforth, for each $v \in V$, let $\mathcal{N}(v) := \{w \in V : \{v, w\} \in E\}$ denote the neighbors of node $v$.

The formulation of the Min-Sum scheme given in Algorithm 1, which we refer to as Min-Sum Splitting, was introduced in [34]. This formulation admits as tuning parameters the real number $\delta \in \mathbb{R}$ and the symmetric matrix $\Gamma = (\Gamma_{vw})_{v,w \in V} \in \mathbb{R}^{V \times V}$. Without loss of generality, we assume that the sparsity of $\Gamma$ respects the structure of the graph $G$, in the sense that if $\{v, w\} \notin E$ then $\Gamma_{vw} = 0$ (note that Algorithm 1 only involves summations with respect to nearest neighbors in the graph). The choice of $\delta = 1$ and $\Gamma = A$, where $A$ is the adjacency matrix defined as $A_{vw} := 1$ if $\{v, w\} \in E$ and $A_{vw} := 0$ otherwise, yields the ordinary Min-Sum algorithm. For

---

**Algorithm 1:** Min-Sum Splitting

---

**Input:** Messages $\hat\mu^0 = (\hat\mu^0_{vw})_{(v,w)\in\mathcal{E}}$; parameters $\delta \in \mathbb{R}$ and $\Gamma \in \mathbb{R}^{V\times V}$ symmetric; time $t \geq 1$.

**for** $s \in \{1,\dots,t\}$ **do**

$\quad\quad \hat\xi^s_{wv} = \phi_v/\delta - \hat\mu^{s-1}_{wv} + \sum_{z\in\mathcal{N}(v)} \Gamma_{zv}\hat\mu^{s-1}_{zv}, (w,v)\in\mathcal{E};$

$\quad\quad \hat\mu^s_{wv} = \min_{z\in\mathbb{R}}\{\phi_{vw}(\,\cdot\,,z)/\Gamma_{vw} + (\delta-1)\hat\xi^s_{wv} + \delta\hat\xi^s_{vw}(z)\}, (w,v)\in\mathcal{E};$

$\mu^t_v = \phi_v + \delta\sum_{w\in\mathcal{N}(v)} \Gamma_{wv}\hat\mu^t_{wv}, v\in V;$

**Output:** $x^t_v = \arg\min_{z\in\mathbb{R}}\mu^t_v(z), v\in V.$

---

an arbitrary choice of strictly positive *integer* parameters, Algorithm 1 can be seen to correspond to the ordinary Min-Sum algorithm applied to a *new formulation* of the original problem, where an equivalent objective function is obtained from the original one in (1) by splitting each term $\phi_{vw}$ into $\Gamma_{vw} \in \mathbb{N}\setminus\{0\}$ terms, and each term $\phi_v$ into $\delta \in \mathbb{N}\setminus\{0\}$ terms. Namely, minimize $\sum_{v\in V}\sum_{k=1}^{\delta}\phi_v^k(x_v) + \sum_{\{v,w\}\in E}\sum_{k=1}^{\Gamma_{vw}}\phi_{vw}^k(x_v,x_w)$, with $\phi_v^k := \phi_v/\delta$ and $\phi_{vw}^k := \phi_{vw}/\Gamma_{vw}$.[1] Hence the reason for the name "splitting" algorithm. Despite this interpretation, Algorithm 1 is defined for any *real* choice of parameters $\delta$ and $\Gamma$.

In this paper we investigate the convergence behavior of the Min-Sum Splitting algorithm for some choices of $\delta$ and $\Gamma$, in the case of the consensus problem that we define in the next section.

## 3 The consensus problem and standard diffusive algorithms

Given a simple graph $G = (V,E)$ with $n := |V|$ nodes, for each $v \in V$ let $\phi_v : \mathbb{R} \to \mathbb{R}\cup\{\infty\}$ be a given function. The consensus problem is defined as follows:

$$\text{minimize}\quad \sum_{v\in V}\phi_v(x_v) \quad \text{subject to}\quad x_v = x_w, \{v,w\}\in E. \tag{2}$$

We interpret $G$ as a communication graph where each node represents an agent, and each edge represent a communication channel between neighbor agents. Each agent $v$ is given the function $\phi_v$, and agents collaborate by iteratively exchanging information with their neighbors in $G$ with the goal to eventually arrive to the solution of problem (2). The consensus problem amounts to designing distributed algorithms to solve problem (2) that respect the communication constraints encoded by $G$.

A classical setting investigated in the literature is the least-square case yielding the network averaging problem, where for a given $b \in \mathbb{R}^V$ we have[2] $\phi_v(z) := \frac{1}{2}z^2 - b_v z$ and the solution of problem (2) is $\bar{b} := \frac{1}{n}\sum_{v\in V}b_v$. In this setup, each agent $v \in V$ is given a number $b_v$, and agents want to exchange information with their neighbors according to a protocol that allows each of them to eventually reach consensus on the average $\bar{b}$ across the entire network. Classical algorithms to solve this problem involve a linear exchange of information of the form $x^t = Wx^{t-1}$ with $x^0 = b$, for a given matrix $W \in \mathbb{R}^{V\times V}$ that respects the topology of the graph $G$ (i.e., $W_{vw} \neq 0$ only if $\{v,w\} \in E$ or $v = w$), so that $W^t \to \mathbf{1}\mathbf{1}^T/n$ for $t \to \infty$, where $\mathbf{1}$ is the all ones vector. This linear iteration allows for a distributed exchange of information among agents, as at any iteration each agent $v \in V$ only receives information from his/her neighbors $\mathcal{N}(v)$ via the update: $x^t_v = W_{vv}x^{t-1}_v + \sum_{w\in\mathcal{N}(v)}W_{vw}x^{t-1}_w$. The original literature on this problem investigates the case where the matrix $W$ has non-negative coefficients and represents the transition matrix of a random walk on the nodes of the graph $G$, so that $W_{vw}$ is interpreted as the probability that a random walk at node $v$ visits node $w$ in the next time step. A popular choice is given by the Metropolis-Hastings method [37], which involved the doubly-stochastic matrix $W^{MH}$ defined as $W^{MH}_{vw} := 1/(2d_{\max})$ if $\{v,w\} \in E$, $W^{MH}_{vw} := 1 - d_v/(2d_{\max})$ if $w = v$, and $W^{MH}_{vw} := 0$ otherwise, where $d_v := |\mathcal{N}(v)|$ is the degree of node $v$, and $d_{\max} := \max_{v\in V} d_v$ is the maximum degree of the graph $G$.

In [44], necessary and sufficient conditions are given for a generic matrix $W$ to satisfy $W^t \to \mathbf{1}\mathbf{1}^T/n$, namely, $\mathbf{1}^T W = \mathbf{1}^T$, $W\mathbf{1} = \mathbf{1}$, and $\rho(W - \mathbf{1}\mathbf{1}^T/n) < 1$, where $\rho(M)$ denotes the spectral radius of a given matrix $M$. The authors show that the problem of choosing the optimal *symmetric* matrix $W$ that minimizes $\rho(W - \mathbf{1}\mathbf{1}^T/n) = \|W - \mathbf{1}\mathbf{1}^T/n\|$ — where $\|M\|$ denotes the spectral norm of a matrix $M$ that coincides with $\rho(M)$ if $M$ is symmetric — is a convex problem and it can be cast as a semi-definite program. Typically, the optimal matrix involves negative coefficients, hence departing from the random walk interpretation. However, even the optimal choice of symmetric matrix is shown to yield a diffusive rate of convergence, which is already attained by the matrix $W^{MH}$ [7]. This rate corresponds to the speed of convergence to stationarity achieved by the diffusion random walk, defined as the Markov chain with transition matrix $\mathrm{diag}(d)^{-1}A$, where $\mathrm{diag}(d) \in \mathbb{R}^{V \times V}$ is the degree matrix, i.e., diagonal with $\mathrm{diag}(d)_{vv} := d_v$, and $A \in \mathbb{R}^{V \times V}$ is the adjacency matrix, i.e., symmetric with $A_{vw} := 1$ if $\{v, w\} \in E$, and $A_{vw} := 0$ otherwise. For instance, the condition $\|W - \mathbf{1}\mathbf{1}^T/n\|^t \le \varepsilon$, where $\|\cdot\|$ is the $\ell_2$ norm, yields a convergence time that scales like $t \sim \Theta(D^2 \log(1/\varepsilon))$ in cycle graphs and tori [33], where $D$ is the graph diameter. The authors in [7] established that for a class of graphs with geometry (polynomial growth or finite doubling dimension) the mixing time of any reversible Markov chain scales at least like $D^2$, and it is achieved by Metropolis-Hastings [37].

## 4 Accelerated algorithms

To overcome the diffusive behavior typical of classical consensus algorithms, two main types of approaches have been investigated in the literature, which seem to have been developed independently.

The first approach involves the construction of a lifted graph $\widehat{G} = (\widehat{V}, \widehat{E})$ and of a linear system supported on the nodes of it, of the form $\hat{x}^t = \widehat{W}\hat{x}^{t-1}$, where $\widehat{W} \in \mathbb{R}^{\widehat{V} \times \widehat{V}}$ is the transition matrix of a *non-reversible* Markov chain on the nodes of $\widehat{G}$. This approach has its origins in the work of [8] and [5], where it was observed for the first time that certain non-reversible Markov chains on properly-constructed lifted graphs yield better mixing times than reversible chains on the original graphs. For some simple graph topologies, such as cycle graphs and two-dimensional grids, the construction of the optimal lifted graphs is well-understood already from the works in [8, 5]. A general theory of lifting in the context of Gossip algorithms has been investigated in [18, 37]. However, this construction incurs additional overhead, which yield non-optimal computational complexity, even for cycle graphs and two-dimensional grids. Typically, lifted random walks on arbitrary graph topologies are constructed on a one-by-one case, exploiting the specifics of the graph at hand. This is the case, for instance, for random geometric graphs [22, 23]. The key property that allows non-reversible lifted Markov chains to achieve subdiffusive rates is the introduction of a directionality in the process to break the diffusive nature of reversible chains. The strength of the directionality depends on global properties of the original graph, such as the number of nodes [8, 5] or the diameter [37]. See Figure 1.

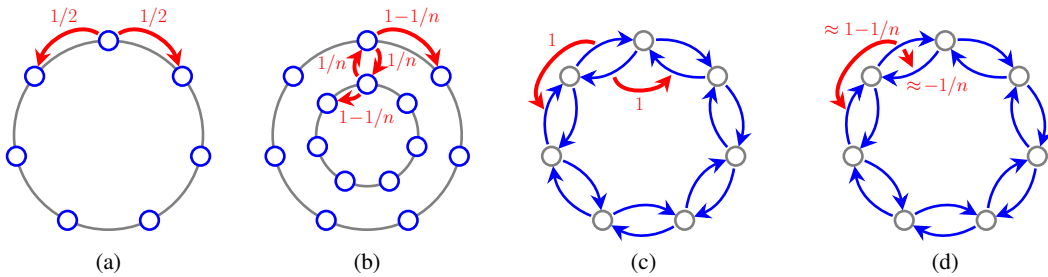

Figure 1: (a) Symmetric Markov chain $W$ on the nodes of the ring graph $G$. (b) Non-reversible Markov chain $\widehat{W}$ on the nodes of the lifted graph $\widehat{G}$ [8]. (c) Ordinary Min-Sum algorithm on the directed edges $\mathcal{E}$ associated to $G$ (i.e., $\widehat{K}(\delta, \Gamma)$, Algorithm 2, with $\delta = 1$ and $\Gamma = A$, where $A$ is the adjacency matrix of $G$). (d) Min-Sum Splitting $\widehat{K}(\delta, \Gamma)$, Algorithm 2, with $\delta = 1$, $\Gamma = \gamma W$, $\gamma = 2/(1 + \sqrt{1 - \rho_W^2})$ as in Theorem 1. Here, $\rho_W$ is $\Theta(1 - 1/n^2)$ and $\gamma \approx 2(1 - 1/n)$ for $n$ large. The matrix $\widehat{K}(\delta, \Gamma)$ has negative entries, departing from the Markov chain interpretation. This is also the case for the optimal tuning in classical consensus schemes [44] and for the ADMM lifting in [12].

The second approach involves designing linear updates that are supported on the original graph $G$ and keep track of a longer history of previous iterates. This approach relies on the fact that the original consensus update $x^t = Wx^{t-1}$ can be interpreted as a primal-dual gradient ascent method to solve problem (2) with a quadratic objective function [32]. This allows the implementation of accelerated

gradient methods. To the best of our knowledge, this idea was first introduced in [14], and since then it has been investigated in many other papers. We refer to [13, 24], and references in there, for a review and comparison of multi-step accelerated methods for consensus. The simplest multi-step extension of gradient methods is Polyak's "heavy ball," which involves adding a "momentum" term to the standard update and yields a primal iterate of the form $x^t = Wx^{t-1} + \gamma(x^{t-1} - x^{t-2})$. Another popular multi-step method involves Nesterov's acceleration, and yields $x^t = (1+\gamma)Wx^{t-1} - \gamma Wx^{t-2}$. Aligned with the idea of adding a momentum term is the idea of adding a shift register term, which yields $x^t = (1+\gamma)Wx^{t-1} - \gamma x^{t-2}$. For our purposes, we note that these methods can be written as

$$
\begin{pmatrix} x^t \\ x^{t-1} \end{pmatrix} = K \begin{pmatrix} x^{t-1} \\ x^{t-2} \end{pmatrix},
\tag{3}
$$

for a certain matrix $K \in \mathbb{R}^{2n \times 2n}$. As in the case of lifted Markov chains techniques, also multi-step methods are able to achieve accelerated rates by exploiting some form of global information: the choice of the parameter $\gamma$ that yields subdiffusive rates depends on the eigenvalues of $W$.

**Remark 1.** *Beyond lifted Markov chains techniques and accelerated first order methods, many other algorithms have been proposed to solve the consensus problem. The literature is vast. As we focus on Min-Sum schemes, an exhaustive literature review on consensus is beyond the scope of our work. Of particular interest for our results is the distributed ADMM approach [3, 43, 38]. Recently in [12], for a class of unconstrained problems with quadratic objective functions, it has been shown that message-passing ADMM schemes can be interpreted as lifting of gradient descent techniques. This prompts for further investigation to connect Min-Sum, ADMM, and accelerated first order methods.*

In the next two sections we show that Min-Sum Splitting bears similarities with both types of accelerated methods described above. On the one hand, in Section 5 we show that the estimates $x_v^t$'s of Algorithm 1 applied to the network averaging problem can be interpreted as the result of a linear process supported on a lifted space, i.e., the space $\mathcal{E}$ of directed edges associated to the undirected edges of $G$. On the other hand, in Section 6 we show that the estimates $x_v^t$'s can be seen as the result of a linear multi-step process supported on the nodes of $G$, which can be written as in (3). Later on, in Section 7 and Section 8, we will see that the similarities just described go beyond the structure of the processes, and they extend to the acceleration mechanism itself. In particular, the choice of splitting parameters that yields subdiffusive convergence rates, matching the asymptotic rates of shift register methods, is also shown to depend on global information about $G$.

## 5 Min-Sum Splitting for consensus

We apply Min-Sum Splitting to solve network averaging. We show that in this case the message-passing protocol is a linear exchange of parameters associated to the directed edges in $\mathcal{E}$.

Given $\delta \in \mathbb{R}$ and $\Gamma \in \mathbb{R}^{V \times V}$ symmetric, let $\hat{h}(\delta) \in \mathbb{R}^{\mathcal{E}}$ be the vector defined as $\hat{h}(\delta)_{wv} := b_w + (1 - 1/\delta)b_v$, and let $\widehat{K}(\delta, \Gamma) \in \mathbb{R}^{\mathcal{E} \times \mathcal{E}}$ be matrix defined as

$$
\widehat{K}(\delta, \Gamma)_{wv,zu} := \begin{cases} \delta \Gamma_{zw} & \text{if } u = w, z \in \mathcal{N}(w) \setminus \{v\}, \\ \delta(\Gamma_{vw} - 1) & \text{if } u = w, z = v, \\ (\delta - 1)\Gamma_{zv} & \text{if } u = v, z \in \mathcal{N}(v) \setminus \{w\}, \\ (\delta - 1)(\Gamma_{wv} - 1) & \text{if } u = v, z = w, \\ 0 & \text{otherwise.} \end{cases}
\tag{4}
$$

Consider Algorithm 2 with initial conditions $\hat{R}^0 = (\hat{R}_{vw}^0)_{(v,w) \in \mathcal{E}} \in \mathbb{R}^{\mathcal{E}}$, $\hat{r}^0 = (\hat{r}_{vw}^0)_{(v,w) \in \mathcal{E}} \in \mathbb{R}^{\mathcal{E}}$.

---

**Algorithm 2:** Min-Sum Splitting, consensus problem, quadratic case

---

**Input:** $\hat{R}^0, \hat{r}^0 \in \mathbb{R}^{\mathcal{E}}$; $\delta \in \mathbb{R}$, $\Gamma \in \mathbb{R}^{V \times V}$ symmetric; $\widehat{K}(\delta, \Gamma)$ defined in (5); $t \geq 1$.

**for** $s \in \{1, \dots, t\}$ **do**

$\quad$ $\hat{R}^s = (2 - 1/\delta)\mathbf{1} + \widehat{K}(\delta, \Gamma)\hat{R}^{s-1}$; $\qquad$ $\hat{r}^s = \hat{h}(\delta) + \widehat{K}(\delta, \Gamma)\hat{r}^{s-1}$;

**Output:** $x_v^t := \frac{b_v + \delta \sum_{w \in \mathcal{N}(v)} \Gamma_{wv} \hat{r}_{wv}^t}{1 + \delta \sum_{w \in \mathcal{N}(v)} \Gamma_{wv} \hat{R}_{wv}^t}$, $v \in V$.

---

**Proposition 1.** *Let $\delta \in \mathbb{R}$ and $\Gamma \in \mathbb{R}^{V \times V}$ symmetric be given. Consider Algorithm 1 applied to problem (2) with $\phi_v(z) := \frac{1}{2}z^2 - b_v z$ and with quadratic initial messages: $\hat{\mu}^0_{vw}(z) = \frac{1}{2}\hat{R}^0_{vw}z^2 - \hat{r}^0_{vw}z$, for some $\hat{R}^0_{vw} > 0$ and $\hat{r}^0_{vw} \in \mathbb{R}$. Then, the messages will remain quadratic, i.e., $\hat{\mu}^s_{vw}(z) = \frac{1}{2}\hat{R}^s_{vw}z^2 - \hat{r}^s_{vw}z$ for any $s \geq 1$, and the parameters evolve as in Algorithm 2. If $1 + \delta \sum_{w \in \mathcal{N}(v)} \Gamma_{wv}\hat{R}^t_{wv} > 0$ for any $v \in V$ and $t \geq 1$, then the output of Algorithm 2 coincides with the output of Algorithm 1.*

## 6 Auxiliary message-passing scheme

We show that the output of Algorithm 2 can be tracked by a *new* message-passing scheme that corresponds to a multi-step linear exchange of parameters associated to the nodes of $G$. This auxiliary algorithm represents the main tool to establish convergence rates for the Min-Sum Splitting protocol, i.e., Theorem 1 below. The intuition behind the auxiliary process is that while Algorithm 1 (hence, Algorithm 2) involves an exchange of messages supported on the directed edges $\mathcal{E}$, the computation of the estimates $x^t_v$'s only involve the *belief functions* $\mu^t_v$'s, which are supported on the nodes of $G$. Due to the simple nature of the pairwise equality constraints in the consensus problem, in the present case a reparametrization allows to track the output of Min-Sum via an algorithm that directly updates the belief functions on the nodes of the graph, which yields Algorithm 3.

Given $\delta \in \mathbb{R}$ and $\Gamma \in \mathbb{R}^{n \times n}$ symmetric, define the matrix $K(\delta, \Gamma) \in \mathbb{R}^{2n \times 2n}$ as

$$K(\delta, \Gamma) := \begin{pmatrix} (1-\delta)I - (1-\delta)\text{diag}(\Gamma \mathbf{1}) + \delta\Gamma & \delta I \\ \delta I - \delta\text{diag}(\Gamma \mathbf{1}) + (1-\delta)\Gamma & (1-\delta)I \end{pmatrix}, \tag{5}$$

where $I \in \mathbb{R}^{V \times V}$ is the identity matrix and $\text{diag}(\Gamma \mathbf{1}) \in \mathbb{R}^{V \times V}$ is diagonal with $(\text{diag}(\Gamma \mathbf{1}))_{vv} = (\Gamma \mathbf{1})_v = \sum_{w \in \mathcal{N}(v)} \Gamma_{vw}$. Consider Algorithm 3 with initial conditions $R^0, r^0, Q^0, q^0 \in \mathbb{R}^V$.

---

**Algorithm 3: Auxiliary message-passing**

---

**Input:** $R^0, r^0, Q^0, q^0 \in \mathbb{R}^V$; $\delta \in \mathbb{R}$, $\Gamma \in \mathbb{R}^{V \times V}$ symmetric; $K(\delta, \Gamma)$ defined in (5); $t \geq 1$.
**for** $s \in \{1, \dots, t\}$ **do**
$$\begin{pmatrix} r^s \\ q^s \end{pmatrix} = K(\delta, \Gamma) \begin{pmatrix} r^{s-1} \\ q^{s-1} \end{pmatrix}; \qquad \begin{pmatrix} R^s \\ Q^s \end{pmatrix} = K(\delta, \Gamma) \begin{pmatrix} R^{s-1} \\ Q^{s-1} \end{pmatrix};$$
**Output:** $x^t_v := r^t_v / R^t_v, v \in V$.

---

**Proposition 2.** *Let $\delta \in \mathbb{R}$ and $\Gamma \in \mathbb{R}^{V \times V}$ symmetric be given. The output of Algorithm 2 with initial conditions $\hat{R}^0, \hat{r}^0 \in \mathbb{R}^{\mathcal{E}}$ is the output of Algorithm 3 with $R^0_v := 1 + \delta \sum_{w \in \mathcal{N}(v)} \Gamma_{wv}\hat{R}^0_{wv}$, $Q^0_v := 1 - \delta \sum_{w \in \mathcal{N}(v)} \Gamma_{wv}\hat{R}^0_{wv}$, $r^0_v := b_v + \delta \sum_{w \in \mathcal{N}(v)} \Gamma_{wv}\hat{r}^0_{wv}$, and $q^0_v := b_v - \delta \sum_{w \in \mathcal{N}(v)} \Gamma_{vw}\hat{r}^0_{vw}$.*

Proposition 2 shows that upon proper initialization, the outputs of Algorithm 2 and Algorithm 3 are equivalent. Hence, Algorithm 3 represents a tool to investigate the convergence behavior of the Min-Sum Splitting algorithm. Analytically, the advantage of the formulation given in Algorithm 3 over the one given in Algorithm 2 is that the former involves two coupled systems of $n$ equations whose convergence behavior can explicitly be linked to the spectral properties of the $n \times n$ matrix $\Gamma$, as we will see in Theorem 1 below. On the contrary, the linear system of $2m$ equations in Algorithm 2 does not seem to exhibit an immediate link to the spectral properties of $\Gamma$. In this respect, we note that the previous paper that investigated Min-Sum schemes for consensus, i.e., [27], characterized the convergence rate of the algorithm under consideration — albeit only in the case of $d$-regular graphs, and upon initializing the quadratic terms to the fix point — in terms of the spectral gap of a matrix that controls a linear system of $2m$ equations. However, the authors only list results on the behavior of this spectral gap in the case of cycle graphs, i.e., $d = 2$, and present a conjecture for $2d$-tori.

## 7 Accelerated convergence rates for Min-Sum Splitting

We investigate the convergence behavior of the Min-Sum Splitting algorithm to solve problem (2) with quadratic objective functions. Henceforth, without loss of generality, let $b \in \mathbb{R}^V$ be given with $0 < b_v < 1$ for each $v \in V$, and let $\phi_v(z) := \frac{1}{2}z^2 - b_v z$. Define $\bar{b} := \sum_{v \in V} b_v / n$.

Recall from [27] that the ordinary Min-Sum algorithm (i.e., Algorithm 2 with $\delta = 1$ and $\Gamma = A$, where $A$ is the adjacency matrix of the graph $G$) does not converge if the graph $G$ has a cycle.

We now show that a proper choice of the tuning parameters allows Min-Sum Splitting to converge to the problem solution in a subdiffusive way. The proof of this result, which is contained in the supplementary material, relies on the use of the auxiliary method defined in Algorithm 3 to track the evolution of the Min-Sum Splitting scheme. Here, recall that $\|x\|$ denotes the $\ell_2$ norm of a given vector $x$, $\|M\|$ denotes the $\ell_2$ matrix norm of the given matrix $M$, and $\rho(M)$ its spectral radius.

**Theorem 1.** *Let $W \in \mathbb{R}^{V \times V}$ be a symmetric matrix with $W\mathbf{1} = \mathbf{1}$ and $\rho_W := \rho(W - \mathbf{1}\mathbf{1}^T/n) < 1$. Let $\delta = 1$ and $\Gamma = \gamma W$, with $\gamma = 2/(1 + \sqrt{1 - \rho_W^2})$. Let $x^t$ be the output at time $t$ of Algorithm 2 with initial conditions $\check{R}^0 = \hat{r}^0 = 0$. Define*

$$K := \begin{pmatrix} \gamma W & I \\ (1-\gamma)I & 0 \end{pmatrix}, \qquad K^\infty := \frac{1}{(2-\gamma)n} \begin{pmatrix} \mathbf{1}\mathbf{1}^T & \mathbf{1}\mathbf{1}^T \\ (1-\gamma)\mathbf{1}\mathbf{1}^T & (1-\gamma)\mathbf{1}\mathbf{1}^T \end{pmatrix}. \qquad (6)$$

*Then, for any $v \in V$ we have $\lim_{t\to\infty} x_v^t = \bar{b}$ and $\|x^t - \bar{b}\mathbf{1}\| \leq \frac{4\sqrt{2n}}{2-\gamma}\|(K - K^\infty)^t\|$. The asymptotic rate of convergence is given by*

$$\rho_K := \rho(K - K^\infty) = \lim_{t\to\infty}\|(K - K^\infty)^t\|^{1/t} = \sqrt{(1 - \sqrt{1-\rho_W^2})/(1 + \sqrt{1-\rho_W^2})} < \rho_W < 1,$$

*which satisfies $\frac{1}{2}\sqrt{1/(1-\rho_W)} \leq 1/(1-\rho_K) \leq \sqrt{1/(1-\rho_W)}$.*

Theorem 1 shows that the choice of splitting parameters $\delta = 1$ and $\Gamma = \gamma W$, where $\gamma$ and $W$ are defined as in the statement of the theorem, allows the Min-Sum Splitting scheme to achieve the asymptotic rate of convergence that is given by the second largest eigenvalue in magnitude of the matrix $K$ defined in (6), i.e., the quantity $\rho_K$. The matrix $K$ is the same matrix that describes shift-register methods for consensus [45, 4, 24]. In fact, the proof of Theorem 1 relies on the spectral analysis previously established for shift-registers, which can be traced back to [15]. See also [13, 24].

Following [27], let us consider the absolute measure of error given by $\|x^t - \bar{b}\mathbf{1}\|/\sqrt{n}$ (recall that we assume $0 < b_v < 1$ so that $\|b\| \leq \sqrt{n}$). From Theorem 1 it follows that, asymptotically, we have $\|x^t - \bar{b}\mathbf{1}\|/\sqrt{n} \lesssim 4\sqrt{2}\rho_K^t/(2 - \gamma)$. If we define the asymptotic convergence time as the minimum time $t$ so that, asymptotically, $\|x^t - \bar{b}\mathbf{1}\|/\sqrt{n} \lesssim \varepsilon$, then the Min-Sum Splitting scheme investigated in Theorem 1 has an asymptotic convergence time that is $O(1/(1-\rho_K)\log\{[1/(1-\rho_K)]/\varepsilon\})$. Given the last bound in Theorem 1, this result achieves (modulo logarithmic terms) a square-root improvement over the convergence time of diffusive methods, which scale like $\Theta(1/(1 - \rho_W)\log 1/\varepsilon)$. For cycle graphs and, more generally, for higher-dimensional tori — where $1/(1 - \rho_W)$ is $\Theta(D^2)$ so that $1/(1-\rho_K)$ is $\Theta(D)$ [33, 1] — the convergence time is $O(D\log D/\varepsilon)$, where $D$ is the graph diameter.

As prescribed by Theorem 1, the choice of $\gamma$ that makes the Min-Sum scheme achieve a subdiffusive rate depends on global properties of the graph $G$. Namely, $\gamma$ depends on the quantity $\rho_W$, the second largest eigenvalue in magnitude of the matrix $W$. This fact connects the acceleration mechanism induced by splitting in the Min-Sum scheme to the acceleration mechanism of lifted Markov chains techniques (see Figure 1) and multi-step first order methods, as described in Section 4.

It remains to be investigated how choices of splitting parameters different than the ones investigated in Theorem 1 affect the convergence behavior of the Min-Sum Splitting algorithm.

## 8   Conclusions

The Min-Sum Splitting algorithm has been previously observed to yield convergence in settings where the ordinary Min-Sum protocol does not converge [35]. In this paper we proved that the introduction of splitting parameters is not only fundamental to guarantee the convergence of the Min-Sum scheme applied to the consensus problem, but that proper tuning of these parameters yields accelerated convergence rates. As prescribed by Theorem 1, the choice of splitting parameters that yields subdiffusive rates involves global type of information, via the spectral gap of a matrix associated to the original graph (see the choice of $\gamma$ in Theorem 1). The acceleration mechanism exploited by Min-Sum Splitting is analogous to the acceleration mechanism exploited by lifted Markov chain techniques — where the transition matrix of the lifted random walks is typically chosen to depend on the total number of nodes in the graph [8, 5] or on its diameter [37] (global pieces of information) — and to the acceleration mechanism exploited by multi-step gradient methods — where the momentum/shift-register term is chosen as a function of the eigenvalues of a matrix supported on the original graph [13] (again, a global information). Prior to our results, this connection seems to have not been established in the literature. Our findings motivate further studies to generalize the acceleration due to splittings to other problem instances, beyond consensus.

## Acknowledgements

This work was partially supported by the NSF under Grant EECS-1609484.

## Footnotes

[1] As mentioned in [34], one can also consider a more general formulation of the splitting algorithm with $\delta \to (\delta_v)_{v\in V} \in \mathbb{R}$ (possibly also with time-varying parameters). The current choice of the algorithm is motivated by the fact that in the present case the output of the algorithm can be tracked by analyzing a linear system on the nodes of the graph, as we will show in Section 5.

[2] In the literature, the classical choice is $\phi_v(z) := \frac{1}{2}\sum_{v\in V}(z - b_v)^2$, which yields the same results as the quadratic function that we define in the main text, as constant terms in the objective function do not alter the optimal point of the problem but only the optimal value of the objective function.

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
