[Supplementary Material]

# Supplementary Material
# Accelerated consensus via Min-Sum Splitting

We present, in order, the proofs of Proposition 1, Proposition 2, and Theorem 1.

*Proof of Proposition 1.* First of all, note that the optimization problem (2) can be casted in the unconstrained formulation of problem (1) upon choosing the hard barrier function: $\phi_{vw}(z, z') := 0$ if $z = z'$ and $\phi_{vw}(z, z') := \infty$ otherwise. With this choice, the minimization inside the definition of the message updates in Algorithm 1 admits the trivial solution $\hat{\mu}^s_{wv} = (\delta - 1)\hat{\xi}^s_{wv} + \delta\hat{\xi}^s_{vw}$. Hence, Algorithm 1 yields the following update for the messages $\hat{\mu}^s = (\hat{\mu}^s_{wv})_{(w,v)\in\mathcal{E}}$:

$$\hat{\mu}^s_{wv} = (1 - 1/\delta)\phi_v - (\delta - 1)\hat{\mu}^{s-1}_{wv} + (\delta - 1)\sum_{z\in\mathcal{N}(v)}\Gamma_{zv}\hat{\mu}^{s-1}_{zv}$$
$$+ \phi_w - \delta\hat{\mu}^{s-1}_{vw} + \delta\sum_{z\in\mathcal{N}(w)}\Gamma_{zw}\hat{\mu}^{s-1}_{zw}. \tag{1}$$

In vector form, this update can be written as $\hat{\mu}^s = \hat{k}(\delta) + \widehat{K}(\delta, \Gamma)\hat{\mu}^{s-1}$, where $\hat{k}(\delta)_{wv} := \phi_w + (1 - 1/\delta)\phi_v$. From the linearity of the message update it follows that if $\phi_v(z) = \frac{1}{2}z^2 - b_v z$ and if we choose the initial messages to be quadratic functions, then the messages at any time $s > 0$ will remain quadratic. Namely, if we adopt the parametrization $\hat{\mu}^0_{vw}(z) = \frac{1}{2}\hat{R}^0_{vw}z^2 - \hat{r}^0_{vw}z$, then we have $\hat{\mu}^s_{vw}(z) = \frac{1}{2}\hat{R}^s_{vw}z^2 - \hat{r}^s_{vw}z$ with the linear and quadratic parameters updated, respectively, according to $\hat{R}^s = (2-1/\delta)\mathbf{1} + \widehat{K}(\delta, \Gamma)\hat{R}^{s-1}$ and $\hat{r}^s = \hat{h}(\delta) + \widehat{K}(\delta, \Gamma)\hat{r}^{s-1}$. The belief function reads $\mu^t_v(z) = \phi_v(z) + \delta\sum_{w\in\mathcal{N}(v)}\Gamma_{wv}\hat{\mu}^t_{wv}(z) = \frac{1}{2}[1 + \delta\sum_{w\in\mathcal{N}(v)}\Gamma_{wv}\hat{R}^t_{wv}]z^2 - [b_v + \delta\sum_{w\in\mathcal{N}(v)}\Gamma_{wv}\hat{r}^t_{wv}]z$. As by assumption $1 + \delta\sum_{w\in\mathcal{N}(v)}\Gamma_{wv}\hat{R}^t_{wv} > 0$, $x^t_v := \arg\min_{z\in\mathbb{R}}\mu^t_v(z) = \frac{b_v + \delta\sum_{w\in\mathcal{N}(v)}\Gamma_{wv}\hat{r}^t_{wv}}{1 + \delta\sum_{w\in\mathcal{N}(v)}\Gamma_{wv}\hat{R}^t_{wv}}$. $\qquad\square$

*Proof of Proposition 2.* Recall from Algorithm 1 the definition of the belief function at time $s$, i.e., $\mu^s_v := \phi_v + \delta\sum_{w\in\mathcal{N}(v)}\Gamma_{wv}\hat{\mu}^s_{wv}$, and let $\mu^s \in \mathbb{R}^V$ be the vector whose $v$-th component is $\mu^s_v$. Let $\chi^s \in \mathbb{R}^V$ be the vector whose $v$-th component is given by the function $\chi^s_v := \phi_v - \delta\sum_{w\in\mathcal{N}(v)}\Gamma_{vw}\hat{\mu}^s_{vw}$. Let $\phi \in \mathbb{R}^V$ be the vector whose $v$-th component is the function $\phi_v$. By taking the summations of update (1) over $w \in \mathcal{N}(v)$ and $v \in \mathcal{N}(w)$, respectively, and by performing the change of variables as prescribed by the definitions of $\mu^s$ and $\chi^s$ (using that $\Gamma$ is symmetric), we get that the functions $\mu^s_v$'s and $\chi^s_v$'s evolve according to the linear system $(\mu^s, \chi^s)^T = K(\delta, \Gamma)(\mu^{s-1}, \chi^{s-1})^T$, where the matrix $K(\delta, \Gamma)$ is defined as in (5). From the linearity of the message updates it follows that if we choose the initial messages to be quadratic functions, then the messages at any time $s > 0$ will remain quadratic. Namely, if we adopt the parametrization $\mu^0_v(z) = \frac{1}{2}R^0_v z^2 - r^0_v z$ and $\chi^0_v(z) = \frac{1}{2}Q^0_v z^2 - q^0_v z$, then $\mu^s_v(z) = \frac{1}{2}R^s_v z^2 - r^s_v z$ and $\chi^s_v(z) = \frac{1}{2}Q^s_v z^2 - q^s_v z$, where the linear and quadratic parameters are updated according to

$$\begin{pmatrix} r^s \\ q^s \end{pmatrix} = K(\delta, \Gamma)\begin{pmatrix} r^{s-1} \\ q^{s-1} \end{pmatrix}, \qquad \begin{pmatrix} R^s \\ Q^s \end{pmatrix} = K(\delta, \Gamma)\begin{pmatrix} R^{s-1} \\ Q^{s-1} \end{pmatrix}.$$

If $R^t_v > 0$ the final estimates read $x^t_v := \arg\min_{z\in\mathbb{R}}\mu^t_v(z) = r^t_v/R^t_v$. $\qquad\square$

*Proof of Theorem 1.* We analyze Algorithm 3 with initial conditions $R^0 = Q^0 = \mathbf{1}$ and $r^0 = q^0 = b$. By Proposition 2, the output of this algorithm coincides with the output of Algorithm 2 with initial

conditions $\hat{R}^0 = \hat{r}^0 = 0$. As $\Gamma = \gamma W$ with $W\mathbf{1} = \mathbf{1}$, we have $\mathrm{diag}(\Gamma\mathbf{1}) = \gamma\mathrm{diag}(\mathbf{1}) = \gamma I$, and the matrix $K(\delta, \Gamma)$ in (5) reads as the matrix $K$ in (6). By the results in [15] (see also [13]), we know that for the choice of $\gamma$ given in the statement of the theorem the following holds:

1. The matrix $K$ has an eigenvalue 1 and all the remaining $2n - 1$ eigenvalues have magnitude strictly less than one.

2. The second largest eigenvalue in magnitude of $K$ is given by the quantity $\rho_K$ defined in the statement of the theorem.

It can be verified that

$$(\mathbf{1}, \mathbf{1})^T K = (\mathbf{1}, \mathbf{1})^T, \qquad K \begin{pmatrix} \mathbf{1} \\ (1 - \gamma)\mathbf{1} \end{pmatrix} = \begin{pmatrix} \mathbf{1} \\ (1 - \gamma)\mathbf{1} \end{pmatrix}.$$

By Lemma 3 in [24], which is a general version of Theorem 1 in [44], we have $\lim_{t\to\infty} W^t = W^\infty$, where $W^\infty$ is defined as in (6). By taking the limit for $t$ that goes to infinity on the two linear systems that define the message updates in Algorithm 3 we get, respectively,

$$\begin{pmatrix} r^\infty \\ q^\infty \end{pmatrix} = K^\infty \begin{pmatrix} r^0 \\ q^0 \end{pmatrix}, \qquad \begin{pmatrix} R^\infty \\ Q^\infty \end{pmatrix} = K^\infty \begin{pmatrix} R^0 \\ Q^0 \end{pmatrix},$$

which yield $r^\infty = \bar{b}R^\infty$, $q^\infty = (1 - \gamma)r^\infty = \bar{b}Q^\infty$, and $R^\infty = \frac{2}{2-\gamma}\mathbf{1}$, $Q^\infty = (1 - \gamma)R^\infty$. Hence, we have $r_v^\infty / R_v^\infty = \bar{b}$. The error decomposition

$$x_v^t - \bar{b} = \frac{r_v^t}{R_v^t} - \frac{r_v^\infty}{R_v^\infty} = \frac{r_v^t}{R_v^t} - \frac{r_v^\infty}{R_v^t} + \frac{r_v^\infty}{R_v^t} - \frac{r_v^\infty}{R_v^\infty} = \frac{1}{R_v^t}(r_v^t - r_v^\infty) + \frac{r_v^\infty}{R_v^t R_v^\infty}(R_v^\infty - R_v^t)$$

yields, using that $R_v^t \geq 1$ and $\bar{b} < 1$, by the triangle inequality for the $\ell_2$ norm $\|\cdot\|$,

$$\|x^t - \bar{b}\mathbf{1}\| \leq \|r^t - r^\infty\| + \|R^t - R^\infty\| \leq 2\max\{\|r^t - r^\infty\|, \|R^t - R^\infty\|\}.$$

We first bound the term for the quadratic parameters. As

$$\begin{pmatrix} R^t - R^\infty \\ Q^t - Q^\infty \end{pmatrix} = (K - K^\infty) \begin{pmatrix} R^{t-1} \\ Q^{t-1} \end{pmatrix} = (K - K^\infty) \begin{pmatrix} R^{t-1} - R^\infty \\ Q^{t-1} - Q^\infty \end{pmatrix},$$

we have

$$\begin{pmatrix} R^t - R^\infty \\ Q^t - Q^\infty \end{pmatrix} = (K - K^\infty)^t \begin{pmatrix} R^0 - R^\infty \\ Q^0 - Q^\infty \end{pmatrix},$$

from which it follows that

$$\|R^t - R^\infty\| \leq \left\| \begin{pmatrix} R^t - R^\infty \\ Q^t - Q^\infty \end{pmatrix} \right\| \leq \|(K - K^\infty)^t\| \left\| \begin{pmatrix} R^0 - R^\infty \\ Q^0 - Q^\infty \end{pmatrix} \right\|.$$

Given that

$$\left\| \begin{pmatrix} R^0 - R^\infty \\ Q^0 - Q^\infty \end{pmatrix} \right\| = \sqrt{\|\mathbf{1} - R^\infty\|_2^2 + \|\mathbf{1} - Q^\infty\|_2^2},$$

with $\|\mathbf{1} - R^\infty\|_2^2 = \|\mathbf{1} - Q^\infty\|_2^2 = \frac{\gamma^2}{(2-\gamma)^2}n$, we get $\|R^t - R^\infty\| \leq \|(K - K^\infty)^t\|\frac{\gamma}{2-\gamma}\sqrt{2n}$. Proceeding analogously for the linear parameters, we find $\|r^t - r^\infty\| \leq \|(K - K^\infty)^t\|\sqrt{\|r^0 - r^\infty\|_2^2 + \|q^0 - q^\infty\|_2^2}$. We have $\|r^0 - r^\infty\| = \|b - \bar{b}R^\infty\| = \|b - \bar{b}\mathbf{1} + \bar{b}\mathbf{1} - \bar{b}R^\infty\| \leq \|b - \bar{b}\mathbf{1}\| + |\bar{b}|\|\mathbf{1} - R^\infty\|$ so that $\|r^0 - r^\infty\| \leq \sqrt{n} + \frac{\gamma}{(2-\gamma)}\sqrt{n} = \frac{2}{(2-\gamma)}\sqrt{n}$. In the same way we get $\|q^0 - q^\infty\| = \|b - \bar{b}Q^\infty\| \leq \frac{2}{(2-\gamma)}\sqrt{n}$. All together, $\|r^t - r^\infty\| \leq \|(K - K^\infty)^t\|\frac{2}{2-\gamma}\sqrt{2n}$. Finally, as $\gamma < 2$ we obtain $\|x^t - \bar{b}\mathbf{1}\| \leq \frac{4}{2-\gamma}\sqrt{2n}\|(K - K^\infty)^t\|$.

It can be checked that for $z \in [0, 1]$ the following inequalities hold

$$1 - 2z \leq \sqrt{\frac{1 - \sqrt{1 - (1 - z^2)^2}}{1 + \sqrt{1 - (1 - z^2)^2}}} \leq 1 - z.$$

Upon choosing $\rho_W = 1 - z^2$, we recover the bounds stated at the end of Theorem 1. $\qquad\square$