[Reviews · NeurIPS 2017]

Reviewer 1



This paper applies an accelerated variant of the min-sum algorithm, called min-sum splitting, to the distributed consensus problem. The paper is very well written, with the contribution clearly placed in the context of the state of the art in the topic. To the best of my knowledge (although I am not an expert on the topic), the results are novel and constitute a qualitative advance. In particular, the paper presents a novel connection between min-sum algorithms and lifted Markov chain techniques. There is a detail which is not clear in the presentation. In page 4, when describing the equivalent objective function that is minimized by the min-sum algorithm to yield the min-sum splitting scheme, the authors write: "...splitting each term $\phi_{vw}$ into $\Gamma_{vw}$ terms, and each term $\phi_v$ into $\delta$ terms,..." However, it is not clear what this means, since $\delta$ and $\Gamma_{vw}$, as introduced on the previous page are real numbers.

Reviewer 2



This paper studies the convergence rate of a so-called min-sum splitting method on the average consensus problem. In general he paper reads fine but the improvement of the result seems not impressive. Detailed comments are as follows. (1) It writes that This rate is optimal for graphs with good expansion properties, such as the complete graph. In this case the convergence time, i.e., the number of iterations required to reach a prescribed level of error accuracy in the… of the dimension of the problem, as…’’. For complete graphs, the linear rate is 0 because everyone converges to the average in 1 step. Also complete graphs are too special to be representative. So for which general category of graphs the complexity does not depend on the dimension (number of nodes)? Which general category of graphs is considered as good? (2) In this paragraph (same as comment 1), the literature review should include ‘’Linear Time Average Consensus on Fixed Graphs and Implications for Decentralized Optimization and Multi-Agent Control’’ by Olshevsky. Its convergence rate should be reported properly (more explanation will be given in comment 8). The reference mentioned here has reached a rather competitive or ever better bound compared the result of the submission. (3) At the top of page 2, for consensus optimization, important references like On the Linear Convergence of the ADMM in Decentralized Consensus Optimization’’ by Shi, Ling, Kun, Wu, and Yin, Optimal algorithms for smooth and strongly convex distributed optimization in networks’’ by Scaman, Bach, Bubeck, Lee, Massoulié should be cited. Also the authors should report the state-of-the-art algorithms for consensus optimization and their corresponding (linear) convergence rates. (4) When discussing lifted graph and Markov chain, this paper ignored a very related paper Markov Chain Lifting and Distributed ADMM’’ by Franca and Bento. (5) The content of the the last paragraph of page 5 is a long known fact. Should refer to Generalized consensus computation in networked systems with erasure links’’ by Rabbat, Nowak, and Bucklew. In the sequel, the connection between those variants and Heavy ball/Nesterov/Polyak is known to the field. (6) There are many important references regarding consensus optimization the authors have ignored. For example, Extra: An exact first-order algorithm for decentralized consensus optimization’’ by Shi, Ling, Wu, and Yin. Fast distributed gradient methods’’ by Jakovetic, J Xavier, and Moura. (7) Proposition 3 seems to be trivial and is a supplementary contribution. (8) The rate has reached by this paper, D log(D/eps), does not seem to have a significant improvement on the rate D log(1/eps) that has been reached by Linear Time Average Consensus on Fixed Graphs and Implications for Decentralized Optimization and Multi-Agent Control (see comment 2). Especially in the worst case scenario (holds for all graphs), D~n, the bound is even worse than that has been achieved in Linear Time Average Consensus….’’. (9) The paperLinear Time Average Consensus…’’ improves the bound through Nesterov’s acceleration. The reviewer suspects that the so-called Auxiliary message-passing scheme’’ proposed by the authors is again a Nestov’s acceleration applied to min-sum algorithm. This is fine but the analysis is done for consensus which boils down to analyzing a linear system and is supposed to be not hard. The contribution of the paper becomes not clear given such situation. (10) The tiny improvement may come from a careful handle on the spectral gap of graphs. Eventually the worst case bound is still O(n) because O(n)=O(D) for the set of all graphs with n nodes. (11) Line 243 of page 6. The graph is simple but the author is using directed edges. This is confusing. (12) Typo at line 220 of page 6. Laplacian—> Lagrangian. After rebuttal: The reviewer is satisfied with the authors' response. But the evaluation score from this reviewer stays the same.

Reviewer 3



In this paper, the authors present an accelerated variant of the Min-Sum message-passing protocol for solving consensus problems in distributed optimization. The authors use the reparametrization techniques proposed in [Ruozzi and Tatikonda, 2013] and establish rates of convergence for the Min-Sum Splitting algorithm for solving consensus problems with quadratic objective functions. The main tool used for the analysis is the construction of an auxiliary linear process that tracks the evolution of the Min-Sum Splitting algorithm. The main contributions of the paper can be summarized as follows: (i) provide analysis for the Min-Sum splitting algorithm using a new proof technique based on the introduction of an auxiliary process, (ii) design a Min-Sum protocol for consensus problems that achieves better convergence than previously established results, and (iii) show the connection between the proposed method, and lifted Markov chains and multi-step methods in convex optimization. The motivation and contributions of the paper are clear. The paper is well written and easy to follow, however, it does contain several typos and grammatical mistakes (listed below). The proofs of Propositions 1 and 2, and Theorem 1 appear to be correct. Typos and Grammatical errors: - Line 34: “…with theirs neighbors…” -> “…with their neighbors…” - Line 174: “double-stochastic” -> “doubly-stochastic” - Line 183: “… can be casted as…” -> “… can be cast as…” - Line 192: “…class of graph with…” -> “…class of graphs with…” - Line 197: “…which seems to…” -> “…which seem to…” - Line 206: “…additional overheads…” -> “…additional overhead…” - Line 225: “…pugging…” -> “…plugging…” - Line 238: “…are seen to…” -> “…are able to…” - Line 240: “…both type of…” -> “…both types of…” - Line 248: “…also seen to…” -> “…also shown to…” - Line 279-280: “…to convergence to…” -> “…to converge to…” - Line 300: “…,which scales like…” -> “…,which scale like…” - Line 302: “…for the cycle,…” -> “…for cycle graphs,…” Other minor comments: - Lines 220 and 221: Do you mean “Lagrangian” and “Lagrange multipliers” instead of “Laplacian” and “Laplace multipliers”? - The authors present 3 algorithms, and the quantities involved are not always explained or described. For example, what is R_{vw} and r_{vw} in Algorithm 2? Also, in Algorithm 2, the quantities \hat{R}^0 and \hat{r}^0 do not appear to be initialized. Moreover, since the auxiliary linear process is key to the analysis and the central idea of the paper, the authors show clearly state which variables correspond to this in Algorithm 3. The paper also appears to be missing several references. More specifically: - Lines 41 and 43: (Sub)gradient methods for consensus optimization. There are several more references that could be included: -- Bertsekas and Tsitsiklis, Parallel and distributed computation: numerical methods, 1989 -- Sundhar Ram Srinivasan et. al., Incremental stochastic subgradient algorithms for convex optimization, 2009 -- Wei Shi, Extra: An exact first-order algorithm for decentralized consensus optimization, 2015 (and, of course, many more) - Line 170: “The original literature…” - Line 229: work by Polyak (Heavy-ball) - Line 232: work by Nesterov It would be interesting and useful if the authors could answer/comment and address in the paper the following: - Although the paper is a theoretical paper, the authors should comment on the practicality of the method, and when such a method should be used as opposed to other distributed methods for consensus optimization. - What are the limitations of the Min-Sum Splitting method? - What is the intuition behind using the auxiliary process in the Min-Sum Splitting method? - The results provided in this paper are for consensus problems with quadratic objective functions. Can this framework be extended to solve more general consensus problems that often arise in Machine Learning? - The authors should also clearly state why such an approach is of interest in the context of Machine Learning and for the Machine Learning community. In summary, this paper is a purely theoretical paper in which the authors establish rates of convergence using a new proof technique and show the connections between their method and well-established methods in the literature. Overall, the ideas presented in this paper are interesting, however, the practicality of the method and intuition behind the results are missing, as well as some justification for the importance of this result for the Machine Learning community.